# Comparative Evaluation of Four Commercially Available Immunoassays for Therapeutic Drug Monitoring of Infliximab and Adalimumab

**DOI:** 10.3390/ijms241210379

**Published:** 2023-06-20

**Authors:** Florian Rissel, Yoann Cazaubon, Syrine Saffar, Romain Altwegg, Mélanie Artasone, Claire Lozano, Thierry Vincent, Alexandre Jentzer

**Affiliations:** 1Department of Immunology, Saint Eloi, Montpellier University Hospital, Montpellier University, 34295 Montpellier, France; f-rissel@chu-montpellier.fr (F.R.);; 2Institute Desbrest of Epidemiology and Public Health, Institut National de la Santé et de la Recherche Médicale (INSERM), Department of Pharmacology and Toxicology, Montpellier University Hospital, Montpellier University, 34090 Montpellier, France; 3Department of Hepato-Gastroenterology, Saint Eloi, Montpellier University Hospital, Montpellier University, 34295 Montpellier, France

**Keywords:** adalimumab, infliximab, antidrug antibodies, therapeutic drug monitoring, immunoassays comparison

## Abstract

Therapeutic drug monitoring (TDM) of anti-TNF-α is an important tool in clinical practice for inflammatory diseases. In this study, we have evaluated the performance of several assays for drug and antidrug antibodies (ADA) measurement in the serum. 50 sera from patients treated with infliximab (IFX) and 49 sera from patients treated with adalimumab (ADAL) were monitored with four immunoassays. We have compared Promonitor, i-Track10^®^, and ez-track1 assays to our gold standard Lisa Tracker^®^ ELISA using Cohen’s kappa, Passing-Bablok, and Bland–Altman analysis. The qualitative analysis evaluated by Cohen’s kappa values found for IFX measurements an “almost perfect” concordance for Promonitor, “moderate” for i-Track10^®^ and “substantial” for ez-Track1. For ADAL, kappa values were “moderate” for all tested methods. For anti-IFX, kappa values were “almost perfect” for Promonitor, “fair” for i-Track10^®^, and “substantial” for ez-Track1. For anti-ADAL, kappa values were “almost perfect” for all three assays. For quantitative analysis of drug measurements, Pearson’s r values were all above 0.9 and Lin’s concordance coefficients of all immunoassays were around 0.80. Performances of the four evaluated immunoassays were acceptable for TDM based on our laboratory experience. Nevertheless, concordance between the four methods for IFX measurement was not perfect and we recommend the use of the same assay for the follow-up of a given patient. The performances of the four immunoassays evaluated were similar and are acceptable for TDM based on our laboratory experience.

## 1. Introduction

Tumor necrosis factor alpha (TNF-α) is a cytokine with pleiotropic effects initially recognized as a necrosis factor. TNF-α binds to the receptors TNFR1 and TNFR2 which initiate signal transduction pathways leading to inflammation and cell death [1]. Physiologically, TNF-α is crucial for a normal immune response. However, the inappropriate or excessive production of TNF-α leads to inflammatory or autoimmune diseases [1]. Therefore, TNF-α has become the target of therapeutic monoclonal antibodies with the aim of blocking inflammatory dysregulation [2]. Thus, anti-TNF-α monoclonal antibodies are broadly used and are efficient in particular for the treatment of rheumatoid arthritis (RA), inflammatory bowel disease (IBD), psoriasis, psoriatic arthritis, ankylosing spondylitis (AS), and juvenile idiopathic arthritis (JIA) [2]. In this article, we will focus more particularly on two anti-TNF-α biotherapies approved by the FDA (Food and Drug Administration) and the EMA (European Medicines Agency) in these pathologies, adalimumab (ADAL) and infliximab (IFX), which are a fully human and a chimeric IgG1 monoclonal antibody respectively [2]. Despite good efficacy, primary nonresponse, loss of response, and adverse events can be related to the development of antidrug antibodies (ADA) with reduced clinical response and an increased incidence of infusion reactions and injection-site reactions [3]. A large meta-analysis found frequent immunizations against these biotherapies with 25.3% of anti-IFX antibodies (95% CI 19.5–32.3) and 14.1% of anti-ADAL antibodies (95% CI 8.6–22.3) [4] that should be taken into account in current medical practice. In a prospective multicenter cohort study with 1650 anti-TNF-α naïve patients with active luminal Crohn’s disease (CD), the authors have concluded that anti-TNF-α treatment failure is common and can be predicted by low drug concentrations mediated in part by immunogenicity [5]. This multivariable analysis showed that the only factor independently associated with primary nonresponse at week 14 and nonremission at week 54 was low IFX or ADAL serum concentration. In detail, primary nonresponse occurred in 23.8% (95% CI 21.4–26.2) at week 14, nonremission at week 54 occurred in 63.1% (95% CI 60.3–65.8), and adverse events curtailed treatment in 7.8% (95% CI 6.6–9.2) [5]. The immunogenicity, which is precocious [5], might be mitigated by early dose optimization, minimizing the loss of response. Several studies have demonstrated a correlation between exposition and clinical outcomes [6,7]. One important aspect of optimizing the therapeutic response to these drugs is monitoring levels in the patient’s bloodstream, particularly the serum-trough level. Monitoring trough provides valuable insights into the drug’s pharmacokinetics, helping clinicians assess if the drug is being maintained in the therapeutic area. To better understand this time-varying concentration–effect relationship where the concentrations required at induction should be higher than during the maintenance phase [8], the TMDD (target-mediated drug disposition) is a specific PK/PD modeling approach that considers the interaction of a drug with its target molecule in the body. In the case of adalimumab and infliximab, which are monoclonal antibodies targeting TNF-α, TMDD modeling has been employed to better understand their pharmacokinetic and pharmacodynamic properties. TMDD models typically incorporate the following key components: free drug compartment influencing pharmacokinetic behavior, target compartment (TNF-α) leads to the pharmacological effects, internalization processes of the drug–target complex, and production/elimination of the TNF-α which can further impact the overall drug interaction. By utilizing these components, TMDD models could help in predicting drug concentrations, target occupancy, and pharmacological effects over time. They provide insights into the optimal dosing regimens, dosing intervals, and target saturation levels required to achieve the desired therapeutic outcomes [9]. This type of model could be useful to improve the use of therapeutic drug monitoring (TDM). Especially for IBD, two therapeutic TDM strategies exist: reactive TDM that should be used for all biologics for both primary nonresponse and secondary loss of response with therapeutic optimization guided by drug and ADA measurements and proactive TDM after induction and at least once during maintenance in order to achieve adequate serum concentrations of biological drugs [10]. Proactive TDM may also be used in de-escalating anti-TNF-α therapy in patients with clinical remission. In economic terms, TDM related to anti-TNF-α therapy seems to result in cost savings in both IBD and RA patients with no negative impact on efficacy compared to routine IFX dose escalation without TDM [11,12]. Concerning proactive TDM, its implementation in clinical practice is controversial due to several randomized controlled trials that failed to demonstrate its benefit compared with the conventional approach [13]. However, recently, Syversen et al. demonstrated the superiority of proactive TDM during the first 3 years of maintenance dose with infliximab for various inflammatory conditions including IBD compared to a conventional approach based on clinical and biological monitoring [14].

Finally, TDM based on clinical data regarding disease activity and especially estimation of pharmacokinetic parameters of the anti-TNF-α might be useful to guide clinical decision-making and to optimize the dose at the individual level [3]. In the aim of performing TDM, several methods have been developed to measure drugs and ADA levels, including enzyme-linked immunosorbent assay (ELISA), radioimmunoassay (RIA), liquid-chromatography-based homogeneous mobility shift assay (HMSA), and chemiluminescence immunoassay (CLIA). Although there is no gold-standard technique, ELISA represents the most commonly used assay for TDM of anti-TNF-α biotherapies and ADA measurements in clinical practice because it has the advantage of being relatively simple and inexpensive. However, ELISA requires several days to report the results for clinicians without random-access flexibility, making it difficult to obtain results in a timely fashion. Consequently, drug administration or therapeutic optimization based on drug concentration or the presence of ADA will be delayed. On the other hand, RIA and HMSA are more expensive and reserved for specialized laboratories. Concerning drug measurement, the correlation between different methods has been shown to be relatively good [15]. With regard to ADA detection, some assays such as first-generation ELISA are “drug sensitive”, meaning that they can only detect ADA in the absence of a detectable drug in the serum. Although all ADA assays are drug sensitive to some degree, as the majority of them rely on the capture of the ADA by drug, next-generation ELISA, as well as RIA and HMSA, are described as “drug tolerant” so that they can measure ADA even in the presence of a drug in the serum. Therefore, the former assays will detect only free ADA while the latter will measure total antidrug antibodies.

Our study aims to evaluate the performance of several commercially available immunoassays for drugs and ADA measurements with an ELISA (Promonitor assays, GRIFOLS™), a chemiluminescence immunoassay (CLIA) (i-Track10^®^, THERADIAG™) and a point-of-care method (POC) based on a time-resolved fluorescence (TRF) (ez-track1, THERADIAG™) in comparison with an ELISA (Lisa Tracker^®^ (LT DS2), THERADIAG™), our gold standard used in the department of Immunology in the Montpellier University Hospital.

## 2. Results

### 2.1. Immunoassays

The immunoassays characteristics used in this study are summarized in Table 1. The measurement ranges for drugs and ADA are substantially equivalent for the four assays carried out in this study. Nevertheless, the use of arbitrary units (AU) for ADA measured by the ELISA Promonitor assays (GRIFOLS™) and ez-Track1 (THERADIAG™), and the capacity for i-Track10^®^ (THERADIAG™) to measure ADA up to 2000 (µg/mL) without any dilution should be noted. The manufacturer’s instructions mentioned no interference with hemolysis, bilirubin, triglyceride, and rheumatoid factors for THERADIAG^TM^ assays and no interference for rheumatoid factor for the GRIFOLS^TM^ assay.

### 2.2. Imprecision

The mean concentrations for each parameter (10 values/parameter) (IFX, ADAL, and ADA) and the imprecision values of LT DS2, our gold standard used in routine, are represented in Table 2. The coefficient of variation (CV) of the intrarun and inter-run imprecisions were acceptable for all the assays (CV < 20%) according to the Food and Drug Administration [16] (FDA) and the European Medicines Agency (EMA) [17] guidelines. Indeed, the intrarun imprecision CV is from 2.6% to 12.2% and the inter-run imprecision CV is from 3.3% to 12.6%.

### 2.3. Qualitative Analysis between LT DS2 and Promonitor DS2/i-Track10^®^/ez-Track1 for IFX, ADAL, Anti-IFX, and Anti-ADAL Measurement

The results of each immunoassay for drug measurement were stratified into three categories according to the standard concentration range [6]: subtherapeutic (<5 µg/mL for IFX and <8 µg/mL for ADAL), maintenance therapeutic (5–10 µg/mL for IFX and 8–12 µg/mL for ADAL), and supratherapeutic (>10 µg/mL for IFX and >12 µg/mL for ADAL) (Table 3). For IFX, kappa values were “almost perfect” for Promonitor (kappa value = 0.904), “moderate” (kappa value = 0.565) for i-Track10, and “substantial” (kappa value = 0.752) for ez-Track1. For ADAL, kappa values were “moderate” for all tested methods: kappa value = 0.455 for Promonitor, kappa value = 0.517 for i-Track10, and kappa value = 0.401 for ez-Track1.

The results of each immunoassay for antidrug antibodies measurements were stratified into two categories: <10 ng/mL and ≥10 ng/mL, assuming that UA/mL is similar to ng/mL (Table 2) with supplier recommendations defining positives antibodies as >10 UA/mL for ez-Track1 and anti-ADAL Promonitor and >5 UA/mL for anti-IFX Promonitor. For anti-IFX, kappa values were “almost perfect” for Promonitor (kappa value = 0.877), “fair” (kappa value = 0.345) for i-Track10, and “substantial” (kappa value = 0.788) for ez-Track1. For anti-ADAL, kappa values were “almost perfect” for the 3 assays: kappa value = 1 for Promonitor, kappa value = 0.936 for i-Track10, and kappa value = 1 for ez-Track1.

### 2.4. Quantitative Analysis between LT DS2 and Promonitor DS2/i-TRACK10^®^/ez-Track1 for IFX, ADAL, Anti-IFX and Anti-ADAL Measurement

A comparative analysis of clinical applicability between LT DS2 and Promonitor DS2/i-TRACK10^®^/ez-Track1 is shown in Figure 1 which reveals that only Promonitor was consistent with no constant or proportional deviation for IFX (Promonitor DS2 = 0.18 (95% CI, −0.014, 1.57) + 0.90 (95% CI, 0.72, 1.00) × LT DS2 (Figure 1A)) and for ADAL (Promonitor DS2 = 0.68 (95% CI, −0.97, 1.61) + 1.08 (95% CI, 0.93, 1.32) × LT DS2 (Figure 1D)). For i-Track10, there was a proportional deviation for IFX (i-Track10 = 0.019 (95% CI, −2.10, 0.84) + 1.31 (95% CI, 1.13, 1.60) × LT DS2 (Figure 1B)) and for ADAL (i-Track10 = 0.55 (95% CI, −0.22, 1.22) + 1.21 (95% CI, 1.11, 1.33) × LTDS2 (Figure 1E). For ez-Track1, there was a constant deviation for IFX (ez-Track1 = −0.66 (95% CI, −1.64, −0.13) + 0.92 (95% CI, 0.77, 1.05) x LTDS2 (Figure 1C)) and a proportional deviation for ADAL (ez-Track1 = 0.16 (95% CI, −1.10, 0.71) + 1.34 (95% CI, 1.18, 1.52) × LTDS2 (Figure 1F)).

Concerning Lin’s concordance coefficient, all coefficients were above 0.80 except for the comparison concerning ADAL for ez-Track POC. A Lin’s concordance coefficient greater than 0.8 was considered as excellent.

Bland–Altman plots revealed mean differences between our “Gold Standard” (LT DS2) and the three other immunoassays tested (Figure 2). For Promonitor, IFX concentrations were quite similar but 0.45% higher (95% CI: −17.9%, 66.1%) compared with LT DS2, with three points outside the limit of agreement (95% CI: −2.7, 3.6; bias 0.45). ADAL concentrations were 18.8% lower on average (95% CI: −63.3, 22.2%), with two points outside the limit of agreement (95% CI: −3.2, 0.932; bias −1.1). For i-TRACK10^®^, IFX concentrations were on average 28.9% lower (95% CI: −71.4%, 13.7%) compared with LT DS2, with two points outside the limit of agreement (95% CI: −6.1, 1.8; bias −2.13). ADAL concentrations were 26.7% lower on average (95% CI: −49.1, 11.2%), with three points outside the limit of agreement (95% CI: −4.2, 0.04; bias −2.1). For ez-Track1, IFX concentrations were 24.1% higher on average (95% CI: −17.9%, 66.1%) compared with LT DS2, with two points outside the limit of agreement (95% CI: −6.1, 1.8; bias −2.13). ADAL concentrations were 28.7% lower on average (95% CI: −72.4, 21.9%), with two points outside the limit of agreement (95% CI: −5.6, 1.2; bias −2.2).

For antidrug antibodies, Passing–Bablok, Lin’s concordance correlation coefficient, and Bland–Altman are presented in Appendix A. Interpretation is not possible due to the number of values available in the calibration range.

## 3. Discussion

Our study evaluated the performance of ELISA (Promonitor assays, GRIFOLS™), CLIA (i-Track10^®^, THERADIAG™), and ez-track1 (THERADIAG™) in comparison with the ELISA LT DS2 (THERADIAG™) we used routinely in the Department of Immunology of the Montpellier University Hospital. This method has been internally validated and is periodically evaluated and validated by external quality controls. Currently, LT DS2 is the most used commercial kit in France. Thus, LT DS2 was used as the reference method for our study.

For IFX measurements, the qualitative analysis evaluated by Cohen’s kappa values found in order of best to worst correlation: Promonitor, ez-Track1, and i-Track10 (0.904, 0.752, and 0.565, respectively). For ADAL, the three methods seemed equivalent (Cohen’s kappa values 0.455, 0.517, and 0.401, respectively) and the concordance with our gold standard was not satisfactory. In our study, 50 samples were analyzed. Nevertheless, additional data with measurements distributed over the entire dosage range could bring more robustness to our study. For the detection of ADA, anti-ADAL antibodies were well identified by the three assays but i-Track10 failed to correctly identify anti-IFX in sera, possibly due to a low number of immunizations. Promonitor has already been identified as an acceptable assay in drug [16] and ADA [17,18] TDM. Ez-Track1 had quite acceptable results even if this assay was not strictly used as a POC because sera were used in the laboratory instead of whole-blood samples in care units. A further validation on whole blood in care units should therefore be preferable even if the supplier’s recommendations indicate that both serum and whole blood can be used with this device. In our study, qualitative analysis of i-Track10^®^ measurements provided a less strong correlation with LT DS2 for IFX, anti-IFX, and anti-ADA than previously shown [19]. This could be explained by a different choice of stratification range. Indeed, we referred to the publication of Adam S Cheifetz et al. for the measurements of the drugs (IFX: 5–10 µg/mL; ADAL: 8–12 µg/mL) [6] rather than older guidelines (IFX: 3–7 µg/mL; ADAL: 5–8 µg/mL) [20,21]. For ADA detection, we focused on the threshold of positivity of our gold standard (10 ng/mL) which seemed the most sensitive and appropriate value to discriminate whether ADA were absent or present. Berger AE et al. used a threshold of 100 ng/mL in order to discriminate low affinity or transient ADAs and high ADA values which would be associated with a decrease of circulating TNF-α and a loss of treatment efficacy but with the risk to underestimate the presence of some ADAs.

Nevertheless, the quantitative analysis indicated a good concordance between the three evaluated assays and our gold standard. Indeed, all serum drug quantifications (except ADAL measured by ez-Track1) had a concerning Lin’s concordance coefficient above 0.80 which is considered by Altman as excellent. For the i-Track10, the Bland–Altman analysis revealed a tendency to overestimate the drug concentrations that was not found elsewhere [19]. For the ez-Track, we could observe the same tendency. The assay with the closest results to our gold standard remains Promonitor, likely because LT DS2 (THERADIAG™) and Promonitor (GRIFOLS™) are both ELISA.

For the ADA, the quantitative analysis has been reported in the Appendix A due to different measurement ranges and units between the assays. The interpretation was not possible because of the small number of positive values related to the few numbers of immunized patients in our cohort. More data need to be collected to compare the different ADA assays.

In our study, we used different methods to measure drugs and ADA in a large number of sera by ELISA, at the patient’s bedside in POC or with an automated method with random access for chemiluminescence. ELISA is a robust test to accurately quantify drugs and ADA with good sensitivity and specificity but requires working in series without flexibility making it difficult to obtain timely results for an efficient and rapid therapeutic drug optimization in case of treatment failure. POC assays give results to clinicians one by one every 15 min and can be adapted to urgent requests but in small numbers. The maintenance of devices and the management of controls and calibrators in clinical departments are often problematic in the implementation of delocalized biology. However, low drug concentration and/or the presence of ADA could be characterized quickly at the time of the consultation and provide valuable information on the mechanism of therapeutic failure useful for a rapid treatment optimization. The automated CLIA is a random-access assay with an acceptable time to result for clinicians (about 35 min to obtain the first result) and the technical handling time is less than that of ELISAs.

To conclude, the four immunoassays evaluated seem acceptable for TDM in clinical practice based on our laboratory experience. Each assay has advantages that are technique dependent: ELISA makes it possible to measure a large number of sera in series, CLIA is a random-access instrument that decreases the time to results for clinicians, and ez-Track1 could rather be used as a POC method in healthcare centers without specialized immunology laboratories. Nevertheless, it is strongly recommended to systematically use the same assay method for the follow-up of a given patient.

## 4. Materials and Methods

### 4.1. Patients and Samples

Samples were obtained from patients treated with IFX or ADAL in Montpellier Hospital or Nimes Hospital between September and November 2022. Blood samples were collected before anti-TNF-α administration. The sera were stored at 4 °C before immunoassays which were performed within 3 weeks. A total of 50 sera from patients treated by IFX were monitored with the 4 immunoassays tested for IFX and anti-IFX and a total of 49 sera from patients treated by ADAL were monitored with the 4 immunoassays tested for ADAL and anti-ADAL. Low and High IMMUNO-TROL^®^ IFX, ADAL, and anti-drug antibodies (THERADIAGTM, Croissy Beaubourg, France) were measured as patient samples and used for imprecision assays.

### 4.2. Immunoassay Method

Four assays were compared in this study: Promonitor ELISA tests (GRIFOLS™, Barcelona, Spain) and Lisa-Tracker^®^ ELISA (THERADIAG™) (LT DS2) using an automated ELISA DS2^®^ analyzer (Dynex Technologies, Chantilly, VA, USA); the chemiluminescence immunoassay (CLIA) i-Track10 from THERADIAG™; and the POC method e-track1 from THERADIAG™, based on a time-resolved fluorescence (TRF). The assays were performed in the Department of Immunology, Saint-Eloi Montpellier University Hospital. Lisa-Tracker^®^ ELISA (THERADIAG™) was used as the gold standard. Internal quality controls were performed at each experiment as current practice in medical biology. The assay procedures were performed according to the manufacturer’s instructions and the specific protocol for each instrument.

### 4.3. Data Analysis

For qualitative analysis, results obtained for LT DS2, Promonitor, i-Track10, and ez-Track1 were evaluated using Cohen’s kappa for agreement of each pair of assays (LT DS2 versus Tested assays). Cohen’s Kappa coefficient was used to measure the level of agreement between two raters or judges who each classified items into predefined categories. It reflects the concordance that is higher the closer its value is to 1. By following the classification of Landis and Koch [22], the agreement interpretation of Kappa results is as follows: <0 poor, 0–0.2 slight, 0.21–0.40 fair, 0.41–0.60 moderate, 0.61–0.80 substantial, and 0.81–1,00 almost perfect.

For quantitative analysis, Passing–Bablok regression, Lin’s concordance coefficient, and Bland–Altmann were adopted to analyze the agreement between LT DS2 (gold standard) and tested methods.

The Passing–Bablok method is a nonparametric, robust method used for comparing and estimating the relationship between two continuous variables (analytical methods). It is particularly useful when analyzing data that may not satisfy the assumptions of traditional linear regression, such as when there are outliers, heteroscedasticity, or non-normal distribution of data. It does not assume a specific functional form for the relationship between variables. Instead, it estimates the line of best fit by comparing the ranks of observations between the two variables. It provides estimates of the slope, intercept, and their confidence intervals. Passing-Bablok regression is especially valuable when analyzing real-world data that may have characteristics that derogate linear regression assumptions.

Lin’s concordance correlation coefficient (ρc) is a measure of agreement or reliability between two sets of continuous measurements. The utility of Lin’s concordance correlation coefficient lies in its ability to capture both the precision (Pearson’s correlation coefficient) and accuracy (Cβ) of the agreement between the two sets of measurements. Cβ is a bias correction factor, a measure of how far a line of best fit is from the identity line: y = x. It takes into account both the systematic differences (bias) and the dispersion (variability) between the measurements. Its interpretation is by no means set in stone. Altman considers the interpretation of ρc > 0.80 as excellent whereas McBride suggests the following ranges: <0.90 poor, 0.90–0.95: moderate, 0.95–0.99: substantial, and >0.99 almost perfect.

Bland–Altman analysis is a statistical technique used to assess the agreement between two quantitative measurements. The Bland–Altman plot displays the difference between the two measurements (*y*-axis) against their mean (*x*-axis), representing the mean difference between the two measurements and the limits of agreement, which are calculated as the mean difference plus or minus two standard deviations of the differences. The agreement was sufficient if the concentration difference was within ±1.96 SD of the mean concentration difference for ≥67% of the sample pairs [23].

Statistical analysis and graphs were performed using Rstudio (version 4.2.0, ggplot, mcr, DescTools, and blandr packages).

### 4.4. Ethics

The Institutional Review Board (IRB) of Montpellier University Hospital has approved the study (IRB Accreditation number: 198711). The approval number assigned by the IRB was IRB-MTP_2022_07_202201163.

## Figures and Tables

**Figure 1 ijms-24-10379-f001:**
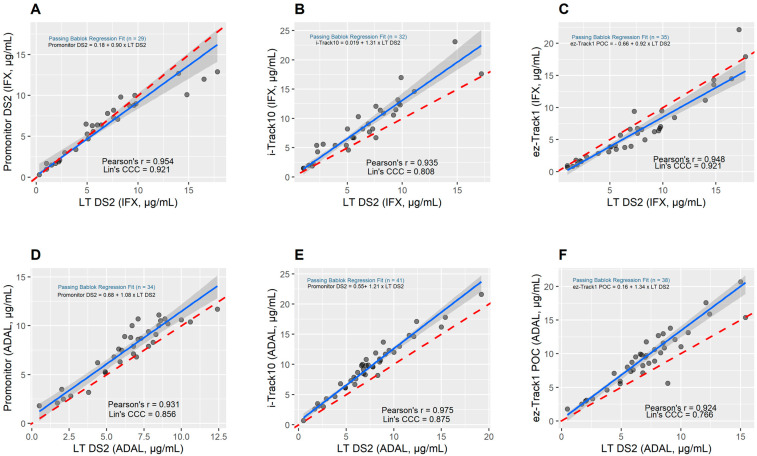
Passing–Bablok regression fit between LT DS2 and Promonitor DS2 (**A**,**D**)/i-TRACK10^®^ (**B**,**E**)/ez-Track1 (**C**,**F**) immunoassays for IFX and ADAL. Pearson’s r-values are shown for all linear correlations. The solid blue lines indicate the Passing–Bablok regression. Red dashed lines are identity lines (y = x). Grey shade areas are the 95% confidence bounds calculated with the bootstrap (quantile) method.

**Figure 2 ijms-24-10379-f002:**
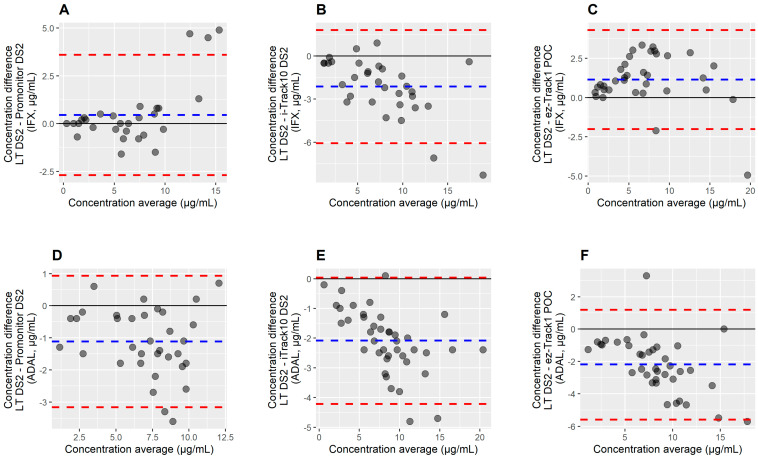
Bland–Altman analysis verifies the difference in IFX/ADAL measurements between LT DS2 and Promonitor DS2 (**A**,**D**)/i-TRACK10^®^ (**B**,**E**)/ez-Track1 (**C**,**F**). The difference between the two measurements is plotted on the *y*-axis and the average of the two measurements on the *x*-axis. Dashed blue lines represent the bias and dashed red lines the 95% limit of agreement (LOA) for each comparison.

**Table 1 ijms-24-10379-t001:** Characteristics of assays used for method comparison.

	Company	Method	Measurement Range			
			IFX	Anti-IFX	ADAL	Anti-ADAL
Promonitor	GRIFOLS^TM^	ELISA (Automated)	0.3–14.4 (µg/mL)	2–144 (AU/mL)	0.25–12 (µg/mL)	6–400 (AU/mL)
LisaTracker^®^	THERADIAG^TM^	ELISA (Automated)	0.3–20 (µg/mL)	10–200 (ng/mL)	0.3–20 (µg/mL)	10–160 (ng/mL)
i-Track^10®^	THERADIAG^TM^	CLIA	0.3–24 (µg/mL)	10–2000 (ng/mL)	0.5–24 (µg/mL)	10–2000 (ng/mL)
ez-track1	THERADIAG^TM^	TRF	0.2–50 (µg/mL)	4–250 (AU/mL)	0.2–50 (µg/mL)	3–200 (AU/mL)

ELISA: enzyme-linked immunoassay CLIA: chemiluminescence immunoassay; TFF: time-resolved fluorescence; IFX: infliximab; ADAL: adalimumab; AU: arbitrary unit.

**Table 2 ijms-24-10379-t002:** Imprecision of LT DS2 with the use of low and high IMMUNO-TROL^®^ IFX, ADAL, and ADA.

	IFX		Anti-IFX		ADAL		Anti-ADAL	
	Intrarun	Inter-Run	Intrarun	Inter-Run	Intrarun	Inter-Run	Intrarun	Inter-Run
	(µg/mL) (CV%)		(ng/mL) (CV%)		(µg/mL) (CV%)		(ng/mL) (CV%)	
Low	4 (4.3)	3.3 (12.6)	41 (4.2)	39.9 (10.1)	3.3 (8.6)	3.5 (11.6)	29 (10.2)	37 (11)
High	9.6 (6.4)	ND	129 (2.6)	ND	11.5 (12.2)	ND	111 (4.8)	ND

IFX: infliximab; ADAL: adalimumab; ADA: antidrug antibodies; CV: coefficient of variation; ND: not determined.

**Table 3 ijms-24-10379-t003:** Data agreement between LT DS2 (gold standard) and Promonitor DS2/i-Track10^®^/ez-Track1 values for IFX, ADAL stratifications according to the therapeutic window in inflammatory bowel diseases during maintenance therapy and for anti-IFX, anti-ADAL stratification according to the limit of quantification (LOQ) of LT DS2 (10 ng/mL).

**IFX**	**LT DS2**				**aIFX**	**LT DS2 (ng/mL)**		
**Promonitor DS2**	**<5 µg/mL**	**5–10 µg/mL**	**>10 µg/mL**	**Total**	**Promonitor DS2 (UA/mL)**	**≥10**	**<10**	**Total**
**i-Track10**					**i-Track10 (ng/mL)**			
**ez-Track1 POC**					**ez-Track1 POC (UA/mL)**			
<5 µg/mL	19	1	0	20	≥10	4	0	4
	17	1	0	18		7	13	20
	21	7	0	28		7	3	10
5–10 µg/mL	1	14	0	15	<10	1	42	43
	4	8	0	12				
	0	11	1	12		0	31	31
>10 µg/mL	0	1	12	13				
	0	10	12	22		0	41	41
	0	0	11	11				
Total	20	16	12	48	Total	5	42	47
	21	19	12	52		7	44	50
	21	18	12	51		7	44	51
Kappa value	0.904				Kappa value	0.877		
	0.565					0.345		
	0.752					0.788		
**ADAL**	**LT DS2**				**aADAL**	**LT DS2 (ng/mL)**		
**Promonitor DS2**	**<8 µg/mL**	**8–12 µg/mL**	**>12 µg/mL**	**Total**	**Promonitor DS2 (UA/mL)**	**≥10**	**<10**	**Total**
**i-Track10**					**i-Track10 (ng/mL)**			
**ez-Track1 POC**					**ez-Track1 POC (UA/mL)**			
<8 µg/mL	26	0	0	26	≥10	8	0	8
	24	0	0	24		9	1	10
	21	1	0	22		8	0	8
8–12 µg/mL	7	9	5	21	<10	0	42	42
	11	7	0	18				
	11	4	0	15		0	43	43
>12 µg/mL	0	2	2	4				
	0	4	7	11		0	41	41
	0	6	7	13				
Total	33	11	7	51	Total	8	42	50
	35	11	7	53		9	44	53
	32	11	7	50		8	41	49
Kappa value	0.455				Kappa value	1.000		
	0.517					0.936		
	0.401					1.000		

## Data Availability

Data are available upon request.

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
