# Peer review of "Comparative Evaluation of Four Commercially Available Immunoassays for Therapeutic Drug Monitoring of Infliximab and Adalimumab"

_ijms, 2023, doi:10.3390/ijms241210379_

Round 1

Reviewer 1 Report

I consider this manuscript of high quality and very useful for our practice. Statistics were fully described and meticulously implemented. I appreciate the attention the authors paid to the details and the way they insisted on commenting and analyzing their findings. The comparison of the three methods to the standard method used in their country gives us a very good idea on what to do and when to use. Discussion paragraph is also excellent, with many nuances in remarks. There are no limitations of the study “per se” (no sub-paragraph), but here and there the authors inserted them, while interpreting their results, with comparison of the literature. I have listed some minor comments for consideration below:

1. After carefully reading the whole manuscript, I do not understand the Conclusion written in the Abstract, given the Results, with some “fair” and “moderate’ concordances. How come was the “performance similar” for all 4 tests, regarding qualitative analysis?

2. Introduction: Line 45: please correct to “primary non-response and loss of response”, instead of “secondary failures”.

3. Figures 1 and 2 (A-F) should be enlarged, especially Fig. 1, in order to be readable and interpreted.

4. Supplementary data were not available for the reviewer.

Minor revision of the English language is required. Occasionally, there are some very long sentences, without comma use.

Reviewer 2 Report

The authors evaluated four immunoassays for TDM of infliximab and adalimumab and concluded that performances of them were similar. Although this study is significant in the clinical settings, outcomes are disputable.

1.       Total number of samples is about 50. For evaluation, the number is too small.

2.       Introduction is too long. Simplify the information that makes up the Introduction. Moreover, four immunoassays need to be explained, especially, Promonitor and Lisa Tracker.

3.       In Discussion, LT DS 2 is used as the gold standard. This misleads readers. Moreover, it is wrong to compare LT DS 2 with others.

Round 2

Reviewer 2 Report

All were revised appropriately.